# Implementing the Integrated Social Sustainability Assessment to Norway: A Citizen-Centric and Expert-Weighted Approach

Tahmineh Akbarinejad *, Alenka Temeljotov Salaj and Agnar Johansen

Department of Civil and Environmental Engineering, Norwegian University of Science and Technology, 7491 Trondheim, Norway
* Correspondence: tahmineh.akbarinejad@ntnu.no; Tel.: +47-48464709

**Abstract:** To achieve sustainability, more economic and environmental initiatives, projects, and policies must have a positive impact on society, advance social justice, and enhance the general well-being of people and communities. This study proposes a quantitative and qualitative framework to assess social sustainability in different urban regions. A multi-category approach is used to determine what categories and indicators of social sustainability city governments and academia should consider to ensure that their policies and projects align with community values. The next step involves assessing residents' satisfaction through citizen participation. This framework, entitled the "Integrated Social Sustainability Assessment (ISSA)", was applied in three zones of the Furuset area in Oslo. The results of the three diagrams show how community strengths and weaknesses can be identified, allowing projects to be prioritized in a way that benefits citizens in the long term and provides a comparative score. This framework provides policymakers with useful tools and guidelines for creating policies and projects that are sustainable, equitable, and capable of meeting the needs of their communities in a measurable manner.

**Keywords:** social sustainability; assessment; framework; neighborhood; indicator; measuring well-being; Norway

## 1. Introduction

In order to achieve climate neutrality in cities, a holistic approach that takes into account social, economic, and environmental factors is required [1,2]. An integrated and strategic approach that takes into account the profound interactions between land use, infrastructure, the built environment, behavior, and policy is required for cities to become more efficient and resilient to climate change [3]. Since the energy transition requires an equitable distribution of benefits and costs among members of society [4], the interpretation and design of the energy transition should involve social sciences and the consideration of analytical, projective, and reflexive perspectives [5].

As part of the United Nations 2030 Agenda, the UN's Sustainable Development Goals have been used since 2015 to assess and monitor the environmental, economic, and social sustainability of nations worldwide [6]. To achieve sustainable development, the three pillars of sustainability must be linked and considered in an integrated and inclusive manner [7]. A lack of consideration for the other pillars can have unintended consequences and undermine sustainability efforts [8–10]. Therefore, social sustainability assessment is essential for identifying potential trade-offs and the unintended consequences of sustainability policies and practices and promoting social equity and inclusion [11]. Cities need to adopt socially inclusive, integrated approaches to sustainable, energy-efficient development to contribute to a more sustainable and equitable future. However, recent studies have shown that, in the majority of well-known assessment methods in developed countries, environmental and economic factors are given more weight than social aspects [12–14]. Because social science is considered inherently complex, it covers a wide

range of issues that usually affect both individuals and society, and these two entities are difficult to distinguish from each other [15]; this challenge needs to be addressed, and there is also a need to develop a comprehensive social sustainability assessment framework to promote dynamic and adaptive approaches to achieve sustainability, which can help stakeholders to anticipate and respond to emerging challenges and opportunities, improve energy efficiency, and promote social and environmental well-being.

In this study, several significant contributions are made to the existing literature on social sustainability assessment and urban development. As a first step, it introduces an innovative Integrated Social Sustainability Assessment (ISSA) tool that integrates citizen and expert perspectives to measure and evaluate social sustainability in neighborhoods. With the ISSA tool, stakeholders' insights are incorporated to ensure a more comprehensive assessment, which addresses the shortcomings of existing methodologies that are heavily focused on economic and environmental aspects. This study addresses this problem by answering the following two research questions:

(1) How can social sustainability be measured by involving citizens and experts?
(2) How can social sustainability criteria and indicators be weighted?

Secondly, this paper presents the implementation of the ISSA tool in the Furuset area of Oslo, demonstrating its practical applicability in evaluating social sustainability in particular urban settings. Real-world applications of the ISSA framework in urban planning demonstrate its potential for guiding policy development and decision-making. Finally, by emphasizing the social dimension of sustainability, this paper fills a critical gap in current assessment frameworks that often overlook the multifaceted and complex nature of social sustainability. Through its citizen-centric and expert-weighted approach, the ISSA tool contributes to fostering just, inclusive, and healthy urban environments, promoting social cohesion, cultural expression, and diversity. Overall, this paper's comprehensive examination and implementation of the ISSA framework could provide valuable insights for researchers, urban planners, and policymakers seeking to advance sustainable and equitable urban development practices, ensuring a more prosperous and harmonious future for cities.

The remainder of this paper is organized as follows: Section 2 contains a review of the existing literature and provides background information on social sustainability and its challenges. In Section 3, the research methodology and main research tools and steps are defined, and a model for the development of the ISSA framework is proposed. Section 4 presents the results derived from applying the framework in the study area. Section 5 discusses the results and answers to the research questions, and Section 6 concludes the study.

## 2. Background

The idea of social sustainability in cities has been raised and debated by a wide spectrum of professionals and groups, and it represents a developing area of study and application that aims to advance just, inclusive, and healthy urban settings.

Social sustainability in cities is promoted through various resources, including the UN's Sustainable Development Goals [16], the New Urban Agenda [17], the Triple Bottom Line [18], and the World Health Organization's Healthy Cities [19] Framework, all of which aim to promote a healthy, equitable, and inclusive environment. However, these frameworks differ in terms of their what they focus on and emphasize. A key aspect of the UN's Sustainable Development Goals is to reduce inequality and guarantee access to basic services while promoting social inclusion and environmental sustainability [20]. The New Urban Agenda emphasizes the role of local governments and stakeholders to create inclusive and equitable cities that enhance social cohesion, cultural expression, and diversity [17], while the Triple Bottom Line, developed by John Elkington in the 1990s, focuses on people, the planet, and profit, attempting to achieve a balance between them and acknowledging the importance of social sustainability in creating inclusive and equitable cities [21]. The New Urban Agenda emphasizes the role of local governments

and stakeholders in creating inclusive and equitable cities that enhance social cohesion, cultural expression, and diversity [22].

Although these resources differ in terms of their focus and approach, they all identify the importance of promoting social sustainability as a crucial aspect of sustainable urban development. By addressing the social, cultural, and economic factors that impact people's lives and promoting equitable and inclusive development, these resources can help create cities that are healthy, vibrant, and sustainable for all residents.

Also, regarding the indicators and trends of social sustainability, historically, social sustainability studies have focused more on macro-scale concepts (cities and regions); however, more recently, the focus of these studies has shifted towards micro-scale concepts (neighborhoods and communities) [23]. Moreover, the traditional "hard" indicators of social sustainability (such as employment and poverty reduction) have given way to more intangible and "soft" measurements [24].

Measuring sustainability is critical for determining the potential environmental impacts of buildings and materials [25]. Numerous sustainability assessment methodologies have been developed globally [26]. These assessment frameworks have established benchmarks for industrialized countries seeking to improve urban areas and residential building sustainability through quantifying environmental performance [13]. Not only are sustainability evaluation frameworks advantageous for buildings but they may also result in healthier inhabitants when social sustainability problems are considered. Recent studies have shown that most sustainability assessment models on the market are focused only on environmental and economic concerns, implying that a social assessment framework is sorely needed [27]. Developing countries have encountered various limitations in implementing such schemes in their buildings. It was discovered that most of these schemes were geared toward the environment and lacked a social aspect to ensure their sustainability [26,28].

Several researchers have recently presented various approaches for evaluating and measuring social sustainability. Compared to other green building assessment techniques like LEED® or BREEM®, few articles have provided a framework for measuring and grading social sustainability in practice. As a result, the goal of this research study was to fill this gap by developing a framework to assess social sustainability. This framework was subsequently applied to the Furuset area in Oslo to evaluate the area's level of social sustainability and evaluate how future projects in this area can improve local social sustainability.

*2.1. The Study Area: Furuset*

The Furuset area of Oslo is located in the Grorud valley within the Oslo city boundaries northwest of the city center. The Grorud Valley is a diverse part of the outskirts of Oslo, with large areas of detached housing, satellite towns, industrial estates, and logistics centres. As part of OBOS (Oslo og omegn Bolig og Sparelag), the satellite town was developed and built in the 1970s. The masterplan included approximately 2800 housing units, schools, nurseries, a care home, a shopping mall, and other commercial properties and public services. Neighboring Furuset are the residential areas of Høybråten and Old Furuset, which are primarily dominated by detached housing. Large green spaces surround the area. Furuset is home to approximately 9000 people from 140 different nationalities. Furuset is a transportation hub, hosting a large amount of daily commuters and travelers. The new development at Furuset is situated next to an existing, established, and effective public transport hub like Trygve Lies Square or Bydelshuset, Fubiak. Moreover, In Furuset, there are some important buildings and areas, including Furuset Church, Furuset Muslim center, IKEA, and some open areas like Furuset kulturpark and parkourpark (Figure 1).

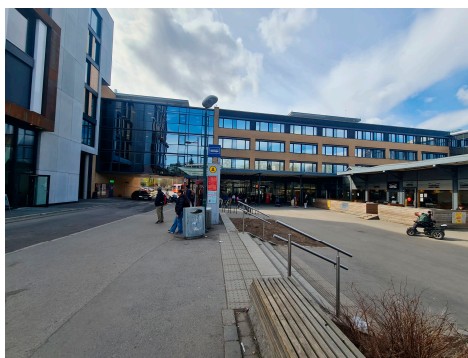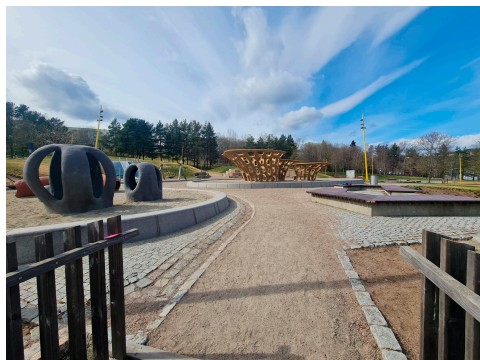

**Figure 1. Left**: Junction of Fubiak, Bydelshu public transport in Trygve square; **Right**: Furuset parkourpark.

This area suffers from a lack of urban qualities; the public spaces are worn out, and there are not enough meeting places. Additionally, unused spaces left after planning (SLOAP) in the neighborhood contributed to a sense of poor urban quality. These urban spaces soon became desolate without regular users or visitors. The lack of activity and sense of responsibility on the part of the residents made the spaces unattractive and unsafe. The Grorud Valley, and Furuset in particular, faced socio-economic issues such as segregation and low housing prices, causing it to develop a bad reputation. In this study, these areas will be evaluated in three different zones: Stjerneblokkveien, Furuset Sentrum, and Gamle Furuset (Figure 2).

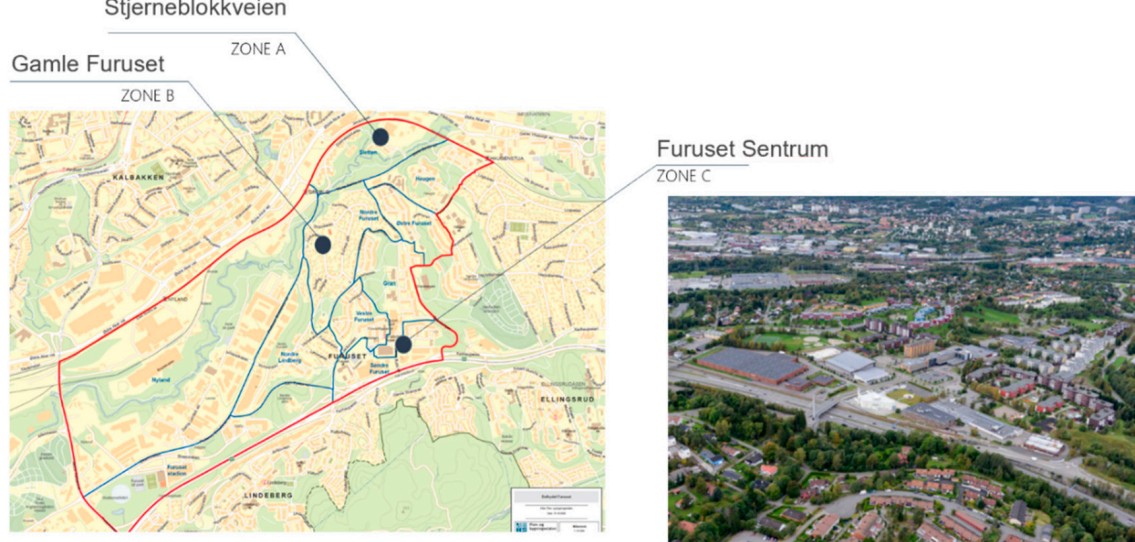

**Figure 2. Left**: the Furuset plan and three different zones; **Right**: bird's-eye view of the Furuset area.

### 2.2. Problem Statement

Due to the necessity of improving social sustainability and the associated challenges ahead, among all of Oslo's regions, the Furuset area has received significant attention. According to the internal reports of the Alna municipality, the main challenges include the following:

- Low average ordinary income among residents aged 30–59 (truly interested members can further join the Plus offering).
- The large amount of unemployed people aged 30–59 (27% compared with 21% in Oslo).
- The fact that the area is a multicultural environment that hosts 140 nationalities.
- The large amount of uneducated people (30.4% for the Alna district).
- A high portion of citizens who did not complete secondary school (over 31%).

- The high population of 16–66 years old with reduced functional ability.

Nonetheless, the Oslo municipality began to focus more on the area in 2007 as part of "Groruddalssatsingen." This initiative is a joint venture between the Oslo municipality and the state which aims to create lasting improvements in certain services and local communities in Groruddalen. The planning procedure began in 2009 by the Planning and Building Services of the City of Oslo and required additional work. The area was selected as one of 50 areas for the FutureBuilt program, which ran between 2010 and 2020. Furuset is a one-of-a-kind FutureBuilt project because it is FutureBuilt's first neighborhood-scale endeavor. The project has a steering group composed of municipal department members. FutureBuilt has selected 50 pilot projects to develop climate-friendly buildings and areas. Many hope that the pilot projects will reduce greenhouse gas emissions from transport, energy, and material consumption by at least 50 percent and inspire changes in the practices of both the private and public sectors. It includes the construction of a school and nursing home and involves upgrading urban areas marked as problematic by participants during the exhibitions and workshops. Norderigo reported that around EUR 1,570,000 (NOK 1.5 billion) has been invested since the Groruddalen action plan was created in 2007 and that around 300 projects have been launched [29].

## 3. Methodology

This section describes the materials and methods used to answer the aforementioned research questions. A comprehensive literature review was conducted by the authors to elucidate the definition of social sustainability [30], and two main questions were answered by introducing the Integrated Social Sustainability Assessment framework (ISSA). As illustrated in Figure 3, the research process was divided into three phases to accomplish the research objectives. For this study, we analyzed quantitative data from an expert panel and qualitative data from residents to optimize the obtained results.

Level 1: Defining research variables, including the main categories and indicators.
Level 2: Evaluating the main categories using an AHP approach while consulting expert opinions.
Level 3: Ranking the case studies' zone(s) in terms of social sustainability using public questionnaires and the proposed Integrated Social Sustainability Assessment Tool (ISSA). At this stage, the residents' level of satisfaction in terms of social sustainability was quantified by using the proposed method. The authors of the present study outlined that, at all levels, interviews with experts were to be conducted to better understand the case studies' needs and challenges through the lens of assessments.

### 3.1. Identified Social Sustainability Indicators

Social sustainability indicators can be used to assess the degree to which a city or urban area is achieving their social sustainability goals. These indicators can be used to track progress over time, identify areas of strength and weakness, and guide the development of policies and strategies designed to promote social sustainability in cities.

Depending on the specific goals and priorities of a particular urban area, there can be many potential indicators of social sustainability. We conducted a comprehensive literature review [30] to identify the main categories and indicators (blue line in the first level). Table 1 shows the different categories identified by different authors in the literature.

**Table 1.** Categories identified by different authors.

| Social Sustainability Categories | Ref. |
| --- | --- |
| Site considerations and equipment, health and comfort considerations, safety and security issues, practitioner interactions, architectural factors. | [26] |
| Social equity, environmental, education, participation and control, social Cohesion, health and safety, accessibility and satisfaction, cultural value, physical resilience. | [31] |
| Equity, education, participation and control, social cohesion, health and safety, accessibility and satisfaction, cultural values. | [32] |
| Social networking and interaction, safety and security, sense of attachment, participation, quality of neighborhoods, quality of housing | [33] |
| Cultural heritage, indoor environment quality, health and well-being, safety and service quality, accessibility, functionality | [34] |
| Health and Comfort, Safety and security, Culture and heritage, Accessibility, Inclusiveness, Participation, Education | [35] |

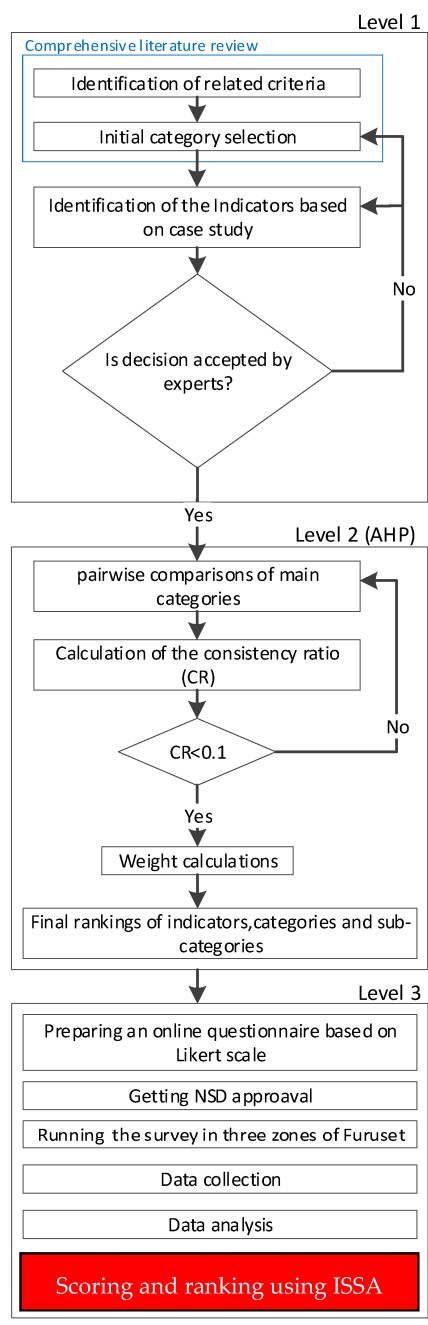

**Figure 3.** Research process flow diagram.

### 3.2. Weighting Categories and Indicators

The AHP method divides the decision-making process into three steps [36]: (1) developing the decision problem's hierarchical pattern, (2) collecting the opinions of experts by applying the AHP fundamental scale (see Table 2), in which numbers are chosen and assigned by experts. Then, a pair-wise comparison matrix is prepared with a 1–9 point scale for decisions, which (3) is used to calculated the Consistency Index (CI). During the third phase, the pair-wise comparison matrix's consistency is computed using CI as shown below:

$$\mathbf{CI} = \frac{\mathbf{\lambda_{max} - n}}{\mathbf{n - 1}} \tag{1}$$

Here, $\mathbf{\lambda_{max}}$ is the eigenvalue, and **n** denotes the number of major criteria (4) calculating the consistency ratio (CR) [37].

$$\mathbf{CR} = \frac{\mathbf{CI}}{\mathbf{RI}} \tag{2}$$

where **RI** is the Random Index obtained from Table 3 [38].

**Table 2.** The fundamental AHP scale (intensity of importance in variables).

| Numeric | Linguistic |
|---|---|
| 1 | Equal importance |
| 3 | Moderate importance |
| 5 | Strong importance |
| 7 | Very strong importance |
| 9 | Extreme importance |
| 2, 4, 6, 8 | Intermediate values between scale values |

**Table 3.** Random Index (RI).

| n | RI | n | RI | n | RI |
|---|---|---|---|---|---|
| 1 | 0.00 | 4 | 0.90 | 7 | 1.32 |
| 2 | 0.00 | 5 | 1.12 | 8 | 1.41 |
| 3 | 0.058 | 6 | 1.24 | 9 | 1.45 |

One critical condition to consider during the pair-wise comparison is that the CR should be less than 0.1. Otherwise, the result would be inaccurate. After completing all steps, the AHP technique calculates the primary criteria and sub-criteria weights [38].

### 3.3. Qualitative Data Collection

To understand the level of inhabitants' satisfaction in each area, questionnaires based on the defined social sustainability categories and indicators were prepared. A 6-point Likert scale methodology was used in the questionnaire. The original Likert scale consists of a series of statements (items) about actual or hypothetical scenarios related to the topic under study [39]. On a metric scale, participants should be asked to indicate their degree of agreement (from strongly disagree to agree strongly) with the supplied statement (items). Combining all the comments reveals the unique character of the approach toward the topic; therefore, the comments are inextricably interconnected [40]. The distinction between Likert-type items and Likert scales was outlined by Clason and Dormody (1994). They determined that Likert-type items are single questions that incorporate some component of the original Likert response alternatives. Multiple questions may be included in a research instrument, but the researcher should not attempt to integrate the responses from the items into a composite scale [41].

Although the 5-point scale is used more frequently than other multiple-choice alternatives [42], after psychology testing, Chomeya (2010) revealed that a 6-point Likert scale can facilitate greater discrimination and reliability than a 5-point Likert scale [43]. The Likert scale is popular in research because it can be easily constructed and modified,

and the results of numerical measurements can be used directly for statistical analysis; Likert-scale-based measurements have good validity, and researchers using a Likert scale can capture large amounts of data with relatively minimal time and effort [42,44].

### 3.4. Data Quantification and Final Scoring

To correctly interpret Likert data (qualitative data), one must comprehend the measuring scale corresponding to each response option. Likert-scale data can be examined using the interval measurement scale. The suggested descriptive statistics for interval scale items are the mean for central tendency and standard deviations for variability. Additional data analysis techniques applicable to interval scale items include Pearson's $\Gamma$, the *t*-test, ANOVA, and regression. For this study, a 6-point Likert scale was used [41]. A 0–5 measuring scale was used to ensure a final scale of 0–100 (Table 4).

**Table 4.** Measuring and Likert scales.

| Measuring Scale | 0 | 1 | 2 | 3 | 4 | 5 |
|---|---|---|---|---|---|---|
| Likert Scale | Strongly Disagree | Disagree | Partly Disagree | Partly Agree | Agree | Strongly Agree |

Among the different methods mentioned above, the central tendency approach was used in the present study. A measure of central tendency is a single number that seeks to characterize data collection by pinpointing its center. Consequently, measurements of central tendency are sometimes referred to as measures of central location [45].

$$SSS = \sum_{i=1}^{34} IW_i \times LSI_i \times 20 \tag{3}$$

where $SSS$ is the final social sustainability score, $IW_i$ is the calculated weight of each indicator in the AHP process, and $LSI_i$ is the Likert scale index, which can be calculated by (4).

$$LSI_i = \frac{\sum_{j=0}^{5} N_j \times j}{\sum_{j=0}^{5} N_j} \tag{4}$$

where $N_j$ is the number of responses to each Likert scale question, and $j$ is the relative measuring scale in Table 5.

**Table 5.** Identified social sustainability categories and their definitions.

|   | Category | Definition |
|---|---|---|
| A | Social equity | Factors related to impartiality, fairness, and justice for all people in social policy. |
| B | Environmental awareness | Factors associated with understanding the natural environment and making choices that benefit Earth rather than hurt it. |
| C | Social cohesion | Factors related to people willingly working and cooperating despite differences in demeanor, culture, and beliefs. |
| D | Health and safety | Factors related to how safe and how healthy the neighborhood is considered among the population that lives there. |
| E | Accessibility | Factors related to commuting and traveling for all people in the neighborhood. |
| F | Cultural value | Factors related to the design of the area and intercultural dialogue between neighbors. |

## 4. Results

### 4.1. Selected Social Sustainability Variables

Among the different categories and indicators mentioned in Section 3.1, six main criteria were selected and extracted from past studies regarding social sustainability assess-

ment methods [30]; the criteria were approved via conducting interviews with members of our expert panel, and their agreement levels were recorded. In order to measure social sustainability in the context of Norway, a set of categories, sub-categories, and indicators were developed based on the literature and the specific characteristics of the community.

Six main categories were established: social equity, environmental awareness, social cohesion, health and safety, accessibility, and cultural value, all of which were evaluated through conducting interviews with four expert panelists (two from the realm of academia and two from the municipality of the study area (Furuset)) to better define each category and the several subcategories and indicators within each category to measure the specific aspects of social sustainability designed by authors. For example, in the category of social equity, equity of process, and fair distribution were considered sub-categories, and this category also contained six indicators, namely access to information, participation in decision-making, the formation of representative groups, the distribution of facilities, chances and opportunities, and equity and non-discrimination. The main sub-categories and indicators are given in Table 6.

**Table 6.** Selected categories and indicators of social sustainability for Norway.

| Category | Sub-Categories | Indicators | |
|---|---|---|---|
| Social Equity (A) | Equity of process | Access to information | A1 |
| | | Participation in decision-making | A2 |
| | | Formation of representative groups | A3 |
| | Fair distribution | Facilities | A4 |
| | | Chances and opportunity | A5 |
| | | Equity and non-discrimination | A6 |
| Environmental Awareness (B) | Environmental awareness and sensibility | Sustainable Materials | B1 |
| | | Clean and renewable energies | B2 |
| | | Water and waste management | B3 |
| | Ecological literacy | Awareness of the physical environment | B4 |
| | | Knowledge of Social events | B5 |
| | | Ability to take action against environmental problems | B6 |
| Social Cohesion (C) | Social programs | Indoor and outdoor social gatherings | C1 |
| | | Neighborhood involvement in design and planning phases | C2 |
| | Social interaction | Design of a place that increases social interaction | C3 |
| | | Sense of belonging | C4 |
| Health and safety (D) | Safety measures | Feeling of safety | D1 |
| | | Relationships between neighbors | D2 |
| | | Street lighting at night | D3 |
| | | Physical resilience in case of hazards | D4 |
| | Health and Indoor Environmental Quality (IEQ) | The physical condition of buildings | D5 |
| | | Clean environment | D6 |
| | | Noise pollution | D7 |
| | | Ventilation | D8 |
| | | Lighting | D9 |
| | | Mental health | D10 |
| | | Life satisfaction | D11 |

**Table 6.** *Cont.*

| Category | Sub-Categories | Indicators | |
|---|---|---|---|
| Accessibility and satisfaction (E) | Ease of accessibility | Access to public transportation | E1 |
| | | Access to public services | E2 |
| | | Accessibility for disabled people | E3 |
| Cultural value (F) | Satisfaction level local identity | Design of building | F1 |
| | | Design of neighborhood | F2 |
| | | Intercultural dialogue | F3 |
| | | Post-occupancy evaluation | F4 |

### 4.2. Weighting Categories and Indicators

In this section, the weights of the social sustainability factors were calculated using the geometric mean AHP approach recommended in the literature [46]. Figure 4 shows the four pair-wise comparison matrices of the main social sustainability categories presented to four experts (two from academia and two from the Alna municipality). Letters A–F correspond to the different categories in Table 6.

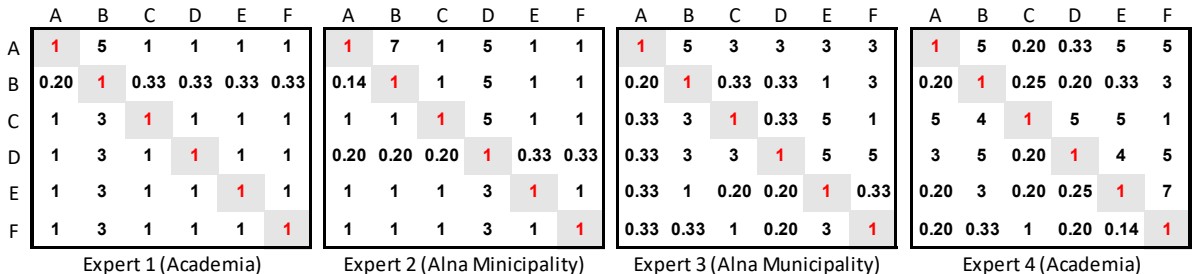

**Figure 4.** Pair-wise comparisons of the main criteria by experts.

Using geometric mean [47], the final averaged pair-wise matrix was calculated as shown in Figure 5. The calculated Consistency Ratio (CR) was 0.039, which is less than 0.1, satisfying the AHP consistency condition.

|   | A | B | C | D | E | F |
|---|---|---|---|---|---|---|
| A | 1 | 5 | 1 | 2 | 2 | 2 |
| B | 0.2 | 1 | 0.5 | 0.5 | 0.5 | 1 |
| C | 1 | 2 | 1 | 2 | 2 | 1 |
| D | 0.5 | 2 | 0.5 | 1 | 2 | 2 |
| E | 0.5 | 2 | 0.5 | 0.5 | 0 | 1 |
| F | 0.5 | 1 | 1 | 0.5 | 1 | 1 |

**Figure 5.** Averaged pair-wise matrix.

The calculated weights and ranking of the six criteria are provided in Figure 6. The results indicate that "social equity" led with 28 percent, followed by "social cohesion" with

22 percent. Experts believe that the weight of these two categories is 50 percent of the total weights. Among all categories, "environmental awareness" received the lowest weight and stood last. The "health and safety" category received 17 percent, and "accessibility and satisfaction" and "cultural value" received 12 percent each.

| | A | B | C | D | E | F | Weight | Rank |
|---|---|---|---|---|---|---|---|---|
| A | 1 | 5 | 1 | 2 | 2 | 2 | 0.28 | 1st |
| B | 0.2 | 1 | 0.5 | 0.5 | 0.5 | 1 | 0.08 | 5th |
| C | 1 | 2 | 1 | 2 | 2 | 1 | 0.22 | 2nd |
| D | 0.5 | 2 | 0.5 | 1 | 2 | 2 | 0.17 | 3rd |
| E | 0.5 | 2 | 0.5 | 0.5 | 1 | 1 | 0.12 | 4th |
| F | 0.5 | 1 | 1 | 0.5 | 1 | 1 | 0.12 | 4th |

**Figure 6.** Weights and rankings of six social sustainability criteria.

By considering the equal weights for the indicators in each category, the final weights of the indicators were calculated, and the results are shown in Table 7.

**Table 7.** Calculated weights of each indicator.

| Indicator | Weight | Indicator | Weight |
|---|---|---|---|
| A1 | 0.0473 | D2 | 0.0157 |
| A2 | 0.0473 | D3 | 0.0157 |
| A3 | 0.0473 | D4 | 0.0157 |
| A4 | 0.0473 | D5 | 0.0157 |
| A5 | 0.0473 | D6 | 0.0157 |
| A6 | 0.0473 | D7 | 0.0157 |
| B1 | 0.0138 | D8 | 0.0157 |
| B2 | 0.0138 | D9 | 0.0157 |
| B3 | 0.0138 | D10 | 0.0157 |
| B4 | 0.0138 | D11 | 0.0157 |
| B5 | 0.0138 | E1 | 0.0406 |
| B6 | 0.0138 | E2 | 0.0406 |
| C1 | 0.0543 | E3 | 0.0406 |
| C2 | 0.0543 | F1 | 0.0305 |
| C3 | 0.0543 | F2 | 0.0305 |
| C4 | 0.0543 | F3 | 0.0305 |
| D1 | 0.0157 | F4 | 0.0305 |

*4.3. Qualitative Data Collection by Survey*

At this stage, the residents' level of satisfaction of 76 local people was collected using the questionnaire described in Section 3.3. The qualitative responses in different zones are given in File S1 in Supplementary Materials. It is possible to rate the three zones of the case study based on the questionnaire results and multiply the indicator's weight using the averages of the Likert scale responses, as shown via the results listed in Table 8. This table provides quantitative data for each indicator, making the zones comparable.

**Table 8.** Social sustainability score (indicator-level).

| Indicators | | Furuset Sentrum | Stjerne Blokkveien | Gamle Furuset |
|---|---|---|---|---|
| Access to information | A1 | 3.04 | 2.95 | 2.96 |
| Participation in decision-making | A2 | 2.26 | 2.73 | 2.52 |
| Formation of representative groups | A3 | 2.16 | 2.65 | 2.17 |
| Facilities | A4 | 2.81 | 3.56 | 2.68 |
| Chances and opportunity | A5 | 3.41 | 3.67 | 2.44 |
| Equity and non-discrimination | A6 | 3.41 | 3.52 | 3.23 |
| Sustainable Materials | B1 | 0.94 | 0.66 | 0.83 |
| Clean and renewable energies | B2 | 0.97 | 0.82 | 0.88 |
| Water and waste management | B3 | 1.06 | 0.89 | 0.86 |
| Awareness of the physical environment | B4 | 1.14 | 0.95 | 0.98 |
| Knowledge of Social events | B5 | 1.01 | 0.91 | 0.88 |
| Ability to take action against environmental problems | B6 | 1.02 | 1.05 | 0.98 |
| Indoor and outdoor social gatherings | C1 | 3.80 | 3.82 | 3.57 |
| Neighborhood involvement in design and'planning | C2 | 3.14 | 3.00 | 3.03 |
| Design of a place that increases social interaction | C3 | 3.61 | 3.13 | 3.03 |
| Sense of belonging | C4 | 4.34 | 4.30 | 3.93 |
| Feeling of safety | D1 | 1.17 | 1.15 | 1.01 |
| Relationships between neighbors | D2 | 1.21 | 1.09 | 1.03 |
| Street lighting at night | D3 | 1.01 | 0.85 | 0.90 |
| Physical resilience in case of hazards | D4 | 1.26 | 1.14 | 0.73 |
| The physical condition of buildings | D5 | 1.20 | 1.09 | 0.80 |
| Clean environment | D6 | 1.25 | 1.14 | 1.00 |
| Noise pollution | D7 | 1.31 | 1.17 | 1.01 |
| Ventilation | D8 | 1.01 | 0.95 | 0.89 |
| Lighting | D9 | 1.23 | 1.04 | 0.87 |
| Mental health | D10 | 1.19 | 1.27 | 0.86 |
| Life satisfaction | D11 | 1.16 | 1.32 | 0.86 |
| Access to public transportation | E1 | 3.42 | 3.22 | 3.11 |
| Access to public services | E2 | 3.31 | 2.66 | 2.98 |
| Accessibility for disabled people | E3 | 2.84 | 2.53 | 2.78 |
| Design of buildings | F1 | 1.89 | 2.05 | 1.68 |
| Design of neighborhood | F2 | 2.13 | 2.14 | 1.83 |
| Intercultural dialogue | F3 | 1.87 | 1.83 | 1.70 |
| Post-occupancy evaluation | F4 | 1.98 | 2.07 | 1.95 |

Using these data, we calculated the social sustainability score of each zone, and the results are shown in Table 9. Moreover, the overall sensitivity score can be graded using Table 10. Based on this system, Gamle Furuset received a Bronze grade, and Furuset Sentrum and Stjerneblokkveien received a Silver grade.

**Table 9.** Social sustainability score (overall and category-level).

| | Furuset Sentrum | Stjerne Blokkveien | Gamle Furuset |
|---|---|---|---|
| Social equity | 17.1 | 19.1 | 16.0 |
| Environmental awareness | 6.1 | 5.3 | 5.4 |
| Social cohesion | 14.9 | 14.2 | 13.6 |
| Health and safety | 13.0 | 12.2 | 10.0 |
| Accessibility and satisfaction | 9.6 | 8.4 | 8.9 |
| Cultural value | 7.9 | 8.1 | 7.2 |
| Overall Score (0–100) | 68.6 | 67.3 | 61.0 |

**Table 10.** Suggested grading system for social sustainability score (based on author suggestions).

| 86–100 | Platinum |
|--------|----------|
| 76–85 | Gold |
| 66–75 | Silver |
| 56–65 | Bronze |
| 46–55 | |
| 45 or less | Reject |

## 5. Discussion

The following discussion is centered around the two main research questions presented in the introduction, with emphasis being placed on the social sustainability framework and its associated characteristics and implementing them by using Furuset as a case study.

(1) *How can social sustainability be measured by involving citizens and experts?*

Several methods and techniques with different approaches were employed in the literature to assess and measure social sustainability in an urban area. However, it is hard to find a practical tool that:

- uses both experts' and residents' opinions;
- compares each social sustainability criterion and indicator;
- illustrates all the information at a glance;
- and gives a final social sustainability score;

The ISSA tool tries to fill these gaps by integrating different quantification methods and calculating the overall score. These scores can result from quantitative and qualitative research in this area. In fact, by illustrating the level of importance of each category based on expert insight and presenting the level of satisfaction among local people, this tool can quantify the score of social sustainability in each zone of the Furuset area. This comparison gives a broad overview of this context's strengths and weaknesses and provides decision-makers with a clear strategy to achieve their goals. By looking at the outcomes of these tools and incorporating all of Furuset's challenges and future criteria, experts can evaluate the present situation and plan for the future of each zone while considering their limitations.

(2) *How can social sustainability criteria and indicators be weighted?*

A key challenge in multi-criteria decision-making is determining the weights of each category [48]. Several weighting methods have been proposed in the literature and used for solving different MCDM problems, such as the Analytic Hierarchy Process (AHP), which has proven successful.

As previously stated, ISSA's main purpose is to develop a framework to measure social sustainability and to provide a dynamic tool based on the experts' opinions and people's satisfaction levels. There is a center circle with multi-layers in the proposed framework. Each layer has the potential for generalization, and development is based on the targets described.

As can be seen in Figure 7, based on Table 5, the first layer focuses on the main criteria of social sustainability as the basis of the tool, and their weights were calculated by averaging the expert responses from the experts from academia and the municipality (Figure 6). The indicators were added to each category based on Table 8, helping to illustrate the second layer. As seen earlier, the main criteria (categories) were weighted using the AHP method. Although this technique could be applied at sub-category- and indicator-level, the indicator's weight in each category was considered to be equal because, at this stage of the study, the main focus of the experts was the evaluation of the weighted main categories based on residents' responses. However, based on the decision-maker's perception of the level of analysis, this step can be included in the future to provide more precise results.

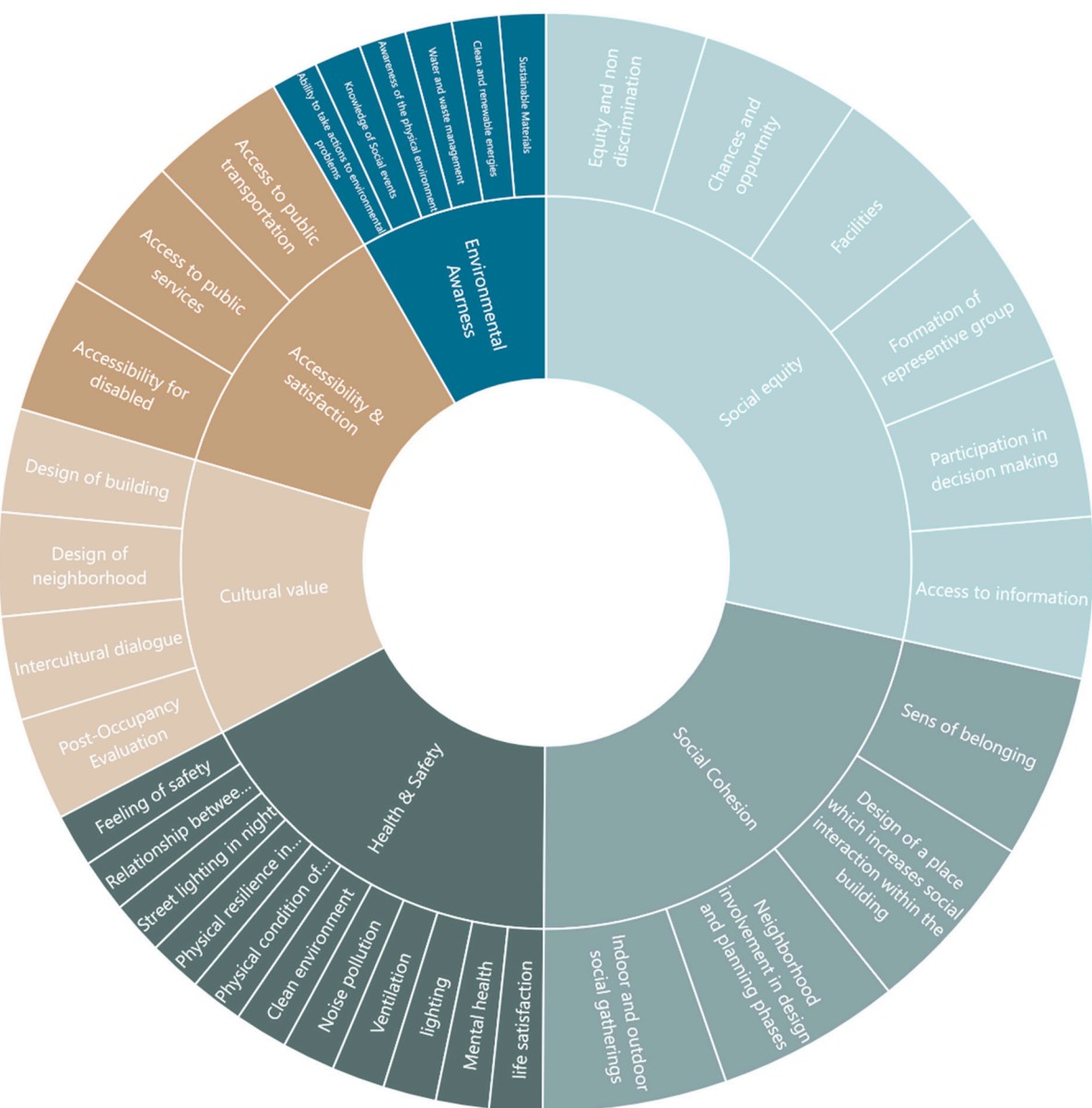

**Figure 7.** Effect of weighting system on indicators in the second level of ISSA tool.

To facilitate a detailed and practical comparison between three zones in the Furuset area and rank them in terms of social sustainability, another layer with the potential to show the quantified people's satisfaction level for each indicator was added to the ISSA tool. These data were collected from the 76 questionnaires using a six-point Likert scale in these three zones, and the data were quantified using the method described in Section 3.4. The green, yellow, and red colors indicate higher, medium, and lower levels of resident satisfaction. The results of the ISSA tool for the three chosen areas can be viewed in Figure 8.

*5.1. Stjerneblokkveien*

The Star-shaped blocks in the Sletten area designed by architects Bernt Heiberg and OM Sandvik were built between 1951 and 1953, and the road was named after them in 1952. Swedish architects Sven Backström and Leif Reinius came up with the idea of a triangular staircase and three protruding arms. The blocks belong to the Stjerneblokkveien housing association, originally called "Jernbanens borettslag" and were built for NSB employees (Figure 9).

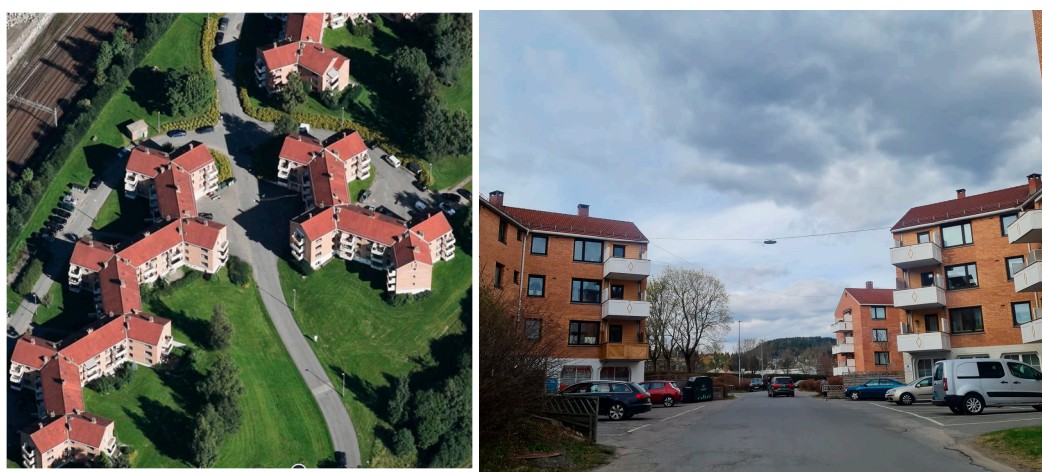

**Figure 8.** ISSA tool results for Stjerneblokkveien.

**Figure 9. Left**: Bird's-eye view of Stjerneblokkveien ("Stjerneblokkveien Borettslag," n.d.). **Right**: Stjerneblokkveien main road.

Based on the survey (File S2 in Supplementary Materials), the ISAA tool illustrates that more than half of the criteria for social sustainability were met.

The main concern is equity in the process of decision-making, which includes participation in decision-making and the formation of representative groups. As shown in Figure 8, if the problem is resolved in the future, this neighborhood is likely to achieve higher scores in these three zones.

Based on the results derived from using the ISSA tool, the citizens of this area tend to get involved in neighborhood projects and need more places to socialize. Providing greater attention to disabled people is considered crucial, and they need equitable facilities relative to other parts of Oslo. Open-ended questions in the survey (File S4 in Supplementary Materials) are consistent with the SSAT tool results.

The majority of complaints pertained to the level of participation in decision-making. Locals seemingly prefer to dedicate votes to all citizens rather than one representative group. Following open questions, it seems a considerable number of the area's occupants are older adults. Many disabled people live in this zone, and they are in need of a store such as a Kiwi or Extra-Coop in their area; in addition, some people have suggested that it would be helpful if carers were assigned to look after them and talk with them. However, overall, cultural value is not well-accepted by citizens. Projects based on improving building design and activities that encourage citizens to embrace intercultural values will enhance their value. In addition, some post-occupancy projects can be carried out in this area to understand their needs better.

In terms of health and safety, the area met expectations. However, the respondents mentioned two areas that require improvement: ventilation and adequate street lighting; improvements in both areas will make the location safer for residents.

There are other values that demonstrate that people should be more educated on sustainability, and there is a clear need for a cleaner environment. The respondents also mentioned that they need more trash bins to keep the area clean. Other comments mainly pertained to children and young people, with respondents mentioning that kids and teenagers need common rooms to stay in in their spare time and that young people need some more activities and opportunities in the area; otherwise, they will waste their time around Furuset Sentrum. A diagram depicting the age ranges of the participants in Stjerneblokkveien is shown in Figure 10.

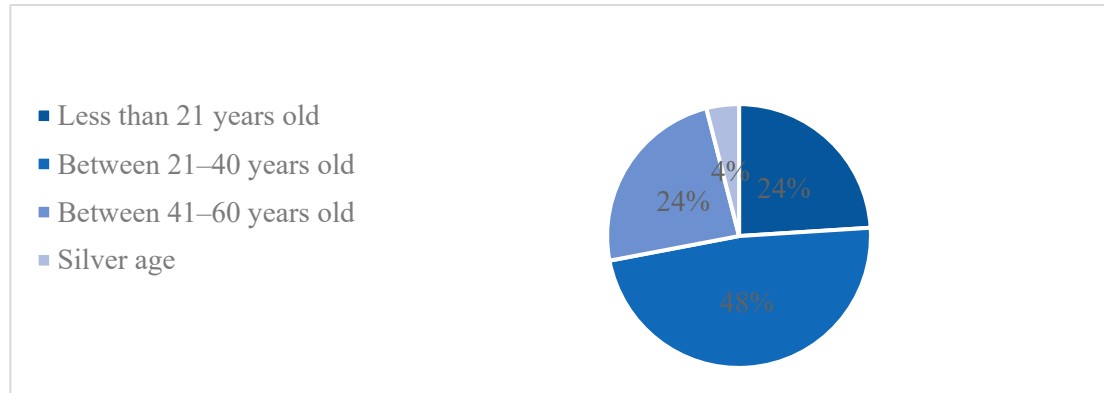

**Figure 10.** Age ranges of participants in Stjerneblokkveien.

*5.2. Furuset Sentrum*

Furuset Sentrum, in this questionnaire, is considered to contain the Gran, Vestre Furuset, and Søndre areas (Figure 11). As a result of implementing ISSA Framework Figure 12 shows that people who are living in these blocks near the Furuset Center and Fublik and Public transportation center are satisfied with nearly half of the indicators of social sustainability in this area.

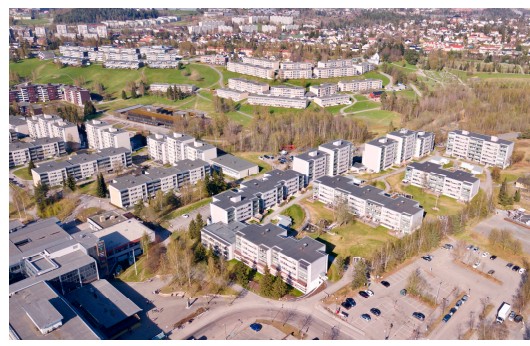
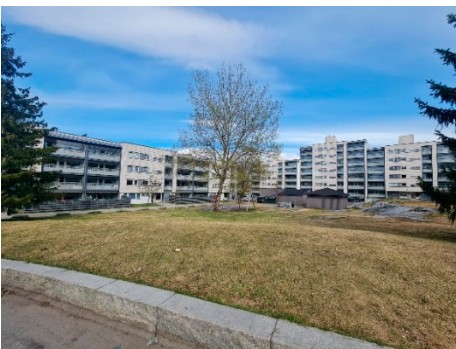

**Figure 11. Left**: Bird's-eye view of Furuset Sentrum. **Right**: View of apartments in Furuset Sentrum.

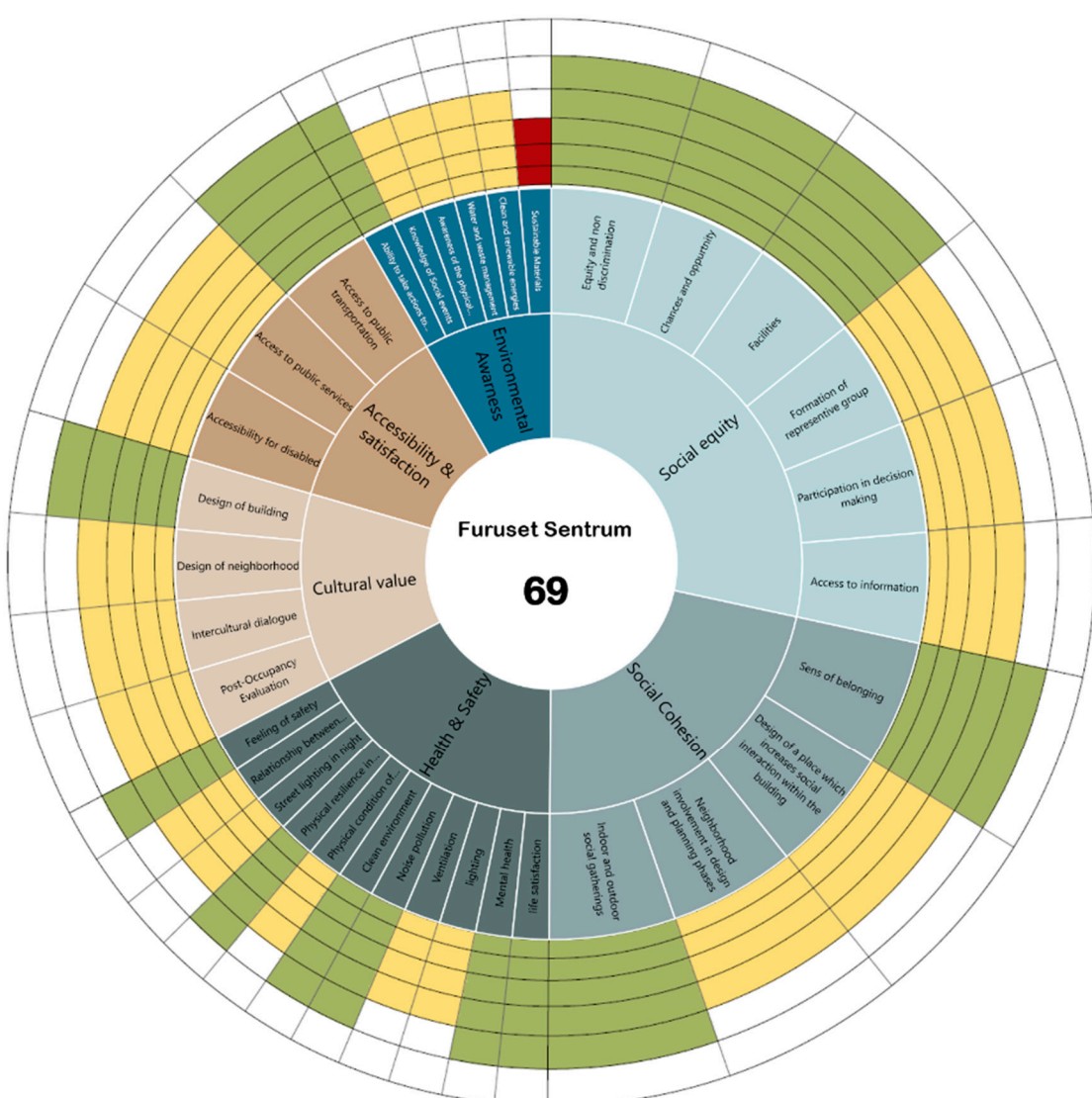

**Figure 12.** ISSA Results in Furuset Sentrum.

However, the data regarding sustainably is poor, and other indicators of environmental awareness indicate that efforts should be made to increase environmental awareness among people in this area.

The results of the ISSA tool indicate that the locals are satisfied with the distribution of facilities in terms of social equity. They need more equity in decision-making processes

and greater accessibility to the information about what is going on in their neighborhood. The same residents in this zone of Stjerneblokkveien also tend to have more social places and participate more in the planning of their communities.

Respondents in this area felt that their neighborhood lacked cultural value. Improving the design of this neighborhood and hosting intercultural events could improve this. Conducting post-occupancy evaluations is another way to increase resident satisfaction levels in this area.

Regarding health and safety, the following improvements are recommended:

- Increasing the amount of natural light in buildings.
- Improving ventilation in buildings.
- The refurbishment and maintenance of materials in their workplaces and common areas.
- Increasing street lighting.
- Social events and celebrations of national holidays to improve relationships between neighbors.
- The residents prefer to have more access to public services, and better conditions are needed for disabled people.

Other changes that could be made to improve residents' quality of life here include improving the conditions of public services and making the public services more accessible for disabled people. Also, municipalities can pay more attention to the cultural value of their neighborhoods by introducing more intercultural spaces and planning for post-evaluations.

Considering the open questions which directly correlate with the results of the SSAT tool, the most interesting thing is that there are both positive and negative responses (File S4 in Supplementary Materials). Most of the answers show that the participants were happy with public transportation and their social environment. Still, some complaints of discrimination against people of different backgrounds or transgender people need to be addressed.

It seems there are also some problems regarding inter-neighbor relations. According to respondents, shared activities in this zone could promote social cohesion and intercultural interaction between neighbors and improve ties.

They also suggested that board members could provide special services for seniors and disabled people who do not live in comfortable conditions in this area. Like Stjerneblockveien, most of the complaints were about providing activities and opportunities for young people to get involved in practical work and enjoy healthily spending their free time. A diagram depicting the age ranges of the participants in Furuset Sentrum can be seen in Figure 13.

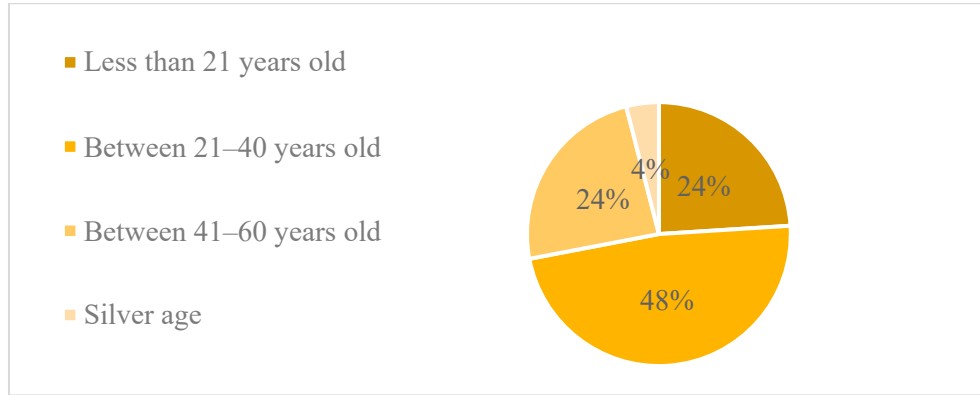

**Figure 13.** Age ranges of participants in Furuset Sentrum.

*5.3. Gamle Furuset*

Gamle Furuset or Nordre Furuset are the oldest parts of Furuset. The Gamle area consists of detached houses built in the 1920s with gardens and vulnerable areas, such as an old hayfield and a stream. According to the people and the municipal authorities

interviewed, their main concern is that many detached houses in old Furuset have been converted into small dormitories, which has caused serious problems (Figure 14).

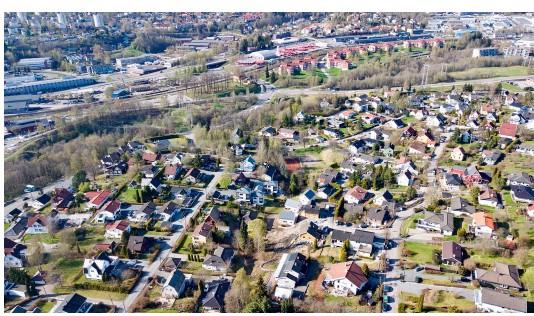 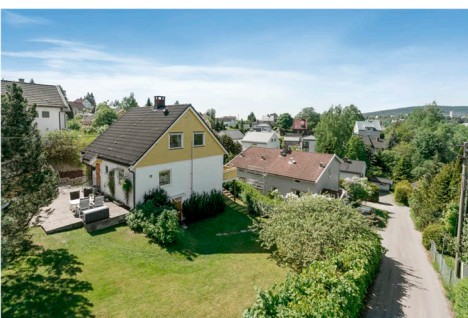

**Figure 14. Left**: Bird's-eye view of Gamle Furuset. **Right**: Detached house with a garden.

As measured by the ISSA tool, the levels social sustainability in this zone are considerably lower than in the other parts and require greater attention. The main problems center around social equity, health and safety, and cultural values (Figure 15). Those who live here feel that they are less likely to find a decent job and enjoy facilities than those in other parts of the Furuset. Respondents also do not think they have a voice in the local government.

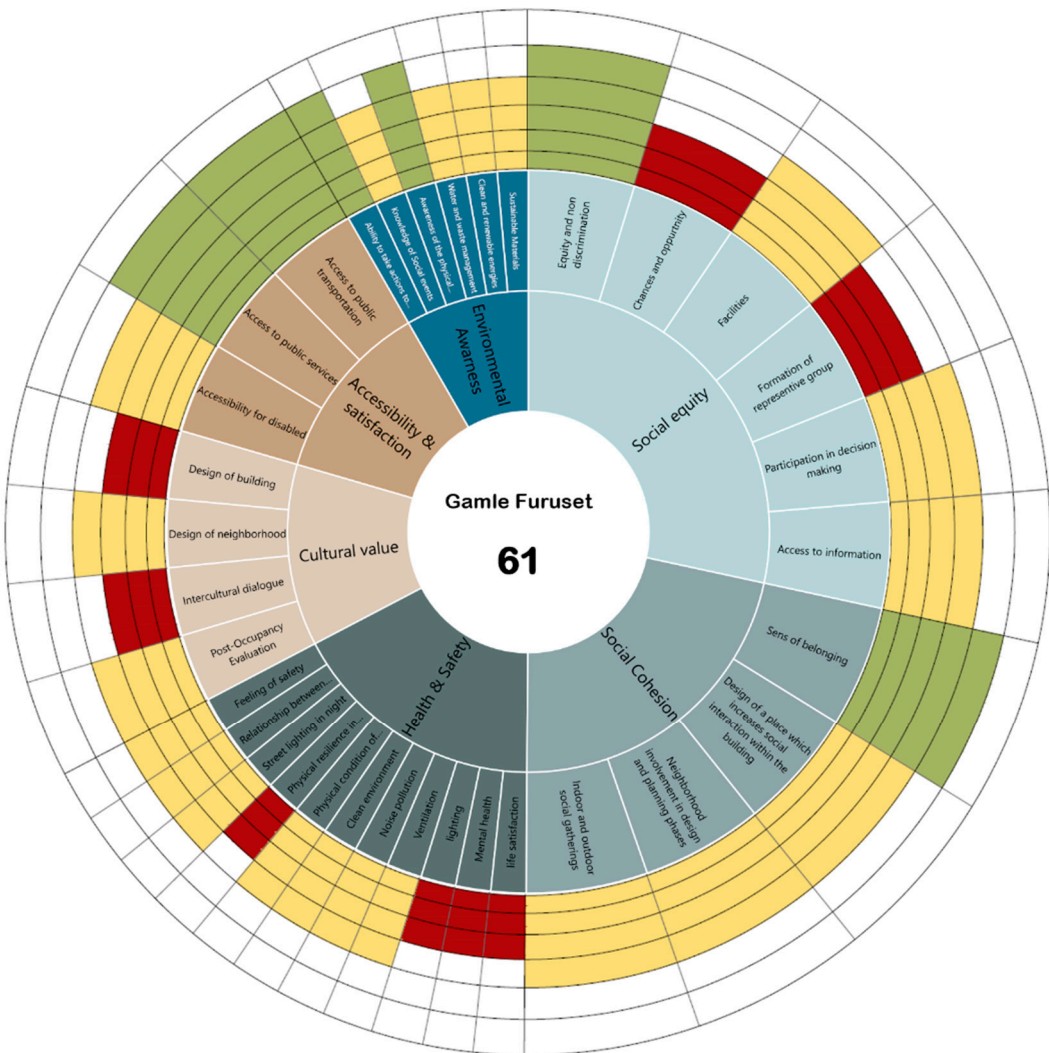

**Figure 15.** ISSA results for Gamle Furuset.

In addition, respondents reported experiencing anxiety and depression. They also feel that the natural lighting in their buildings is insufficient and are not confident about the resilience of their homes in case of disaster. Moreover, they are not satisfied with their buildings and have trouble with engaging in intercultural dialogue with neighbors.

However, despite these problems, they feel a strong sense of belonging to this area. Also, they are satisfied with their access to public services and public transport and do not feel discriminated against; indicators that are colored yellow in Figure 14 are in need of improvement.

Concerning the open questions (File S4 in Supplementary Materials), fewer people provided suggestions and comments in Gamle Furuset compared to the two other zones. In the open questions, There are some very positive feedback was provided with respect to social cohesion and social equity in this area; however, complaints were made about the state of old houses and public buildings. More regular maintenance of houses and roads, blocks and walls, and playgrounds and public buildings is needed.

In addition to prioritizing children's needs, similar to the requests made within other parts of Furuset, there were requests for the provision of more activities and opportunities for young people in this area to prevent problems relating to violence among young people. A diagram depicting the age ranges of the participants in Gamle Furuset can be seen in Figure 16.

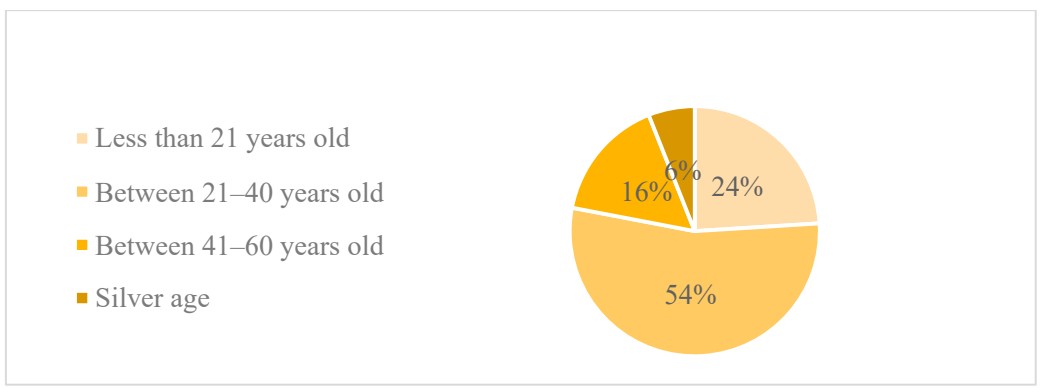

**Figure 16.** Age ranges of participants in Gamle Furuset.

## 6. Conclusions

In conclusion, this study has made a significant contribution to the assessment of social sustainability at the neighborhood scale, filling a knowledge gap in this area. The development of the Integrated Social Sustainability Assessment Tool (ISSA) addresses the limitations of existing methods by providing decision-makers with a reliable and comprehensive tool that combines stakeholder opinions, enhances the understanding of social sustainability parameters in neighborhoods, and offers a clear visualization of the results.

After conductive a comprehensive scoping review, a comprehensive list of social sustainability criteria and indicators was prepared, and a panel of experts selected the most relevant ones. The first layer of the ISSA tool contains categories covering essential aspects such as social equity, environmental awareness, social cohesion, health and safety, accessibility, and cultural value. The use of the Analytical Hierarchy Process (AHP) to weigh the different categories ensured a robust and widely accepted approach for prioritizing different parameters.

The second layer of the ISSA consists of indicators with similar values within each category, facilitating a more nuanced assessment. For example, the category of social equity includes indicators such as access to information, participation in decision-making, the formation of representative groups, facilities, and chances and opportunities. Additionally, the tool's implementation through a survey of resident satisfaction using a six-point Likert scale allowed us to visually represent the level of satisfaction with various indicators.

The development and application of the Integrated Social Sustainability Assessment (ISSA) tool in the Furuset area of Oslo provided valuable insights into social sustainability assessment at the neighborhood scale. However, there are certain limitations that should be acknowledged to contextualize the findings of this study, and caution is needed when generalizing the findings of this study to other urban areas due to Furuset's unique characteristics. Data availability and gaps may have influenced result accuracy, and we urge researchers to explore innovative data collection methods. The selection of indicators involved expert judgment and subjectivity, suggesting the potential for more participatory approaches in future research for a more inclusive assessment. Future research and development in social sustainability assessment should focus on several key directions.

- Rigorously validating and testing the ISSA tool will be crucial for assessing its reliability and validity. Insights can be gained from comparing longitudinal studies with real-world social sustainability results.
- By integrating new data sources like geospatial data, social media analytics, and crowdsourced information through the web base, the tool will be able to provide comprehensive and real-time assessments.
- To address the dynamic nature of urban environments, an adaptive framework with a feedback loop and real-time data updates can enable the continuous improvement of the ISSA tool's assessment process.
- Conducting comparative studies across different cities or regions will reveal patterns, best practices, and learned lessons. This will provide policymakers with a deeper understanding of the factors influencing social sustainability.
- Interdisciplinary collaboration between researchers from different fields could help to enhance the design of the ISSA tool and address complex urban problems.

The ISSA tool's successful integration of different quantification methods and calculation of an overall score offers a comprehensive and practical approach for assessing social sustainability in urban areas. It empowers decision-makers to make informed choices that foster inclusive and equitable neighborhoods, promote social cohesion, and enhance residents' well-being.

The ISSA tool is useful for exploring and communicating complex data in an intuitive and engaging way. It helps decision-makers gain insight into social sustainability concerns, such as the following, at a glance:

- Recognizing the current status of each neighborhood concerning social sustainability for urban planners and architects.
- Identifying zones with weaknesses to look for more result-oriented strategies.
- Evaluating future projects in terms of social sustainability.
- Outlining measurable goals to achieve the right SDGs in this area.
- Determining the most urgent short-term and long-term projects for achieving sustainability goals while considering social challenges.
- Monitoring progress in terms of social sustainability development.
- Increasing value for cities and citizens and ensuring the well-being of individuals.

The framework can be applied in a wide range of contexts and on many different levels. The social sustainability assessment, as is the case with many other real-life applications, suffers from ambiguity, and it has considerable natural uncertainties. To make the tool more robust for future studies, we suggest implementing uncertainty modeling techniques such as fuzzy set theory into all levels. Moreover, for future case studies, it would be beneficial to use a broader expert panel and employ the Delphi technique for interviews. Accordingly, variables such as categories and indicators can be adapted depending on how the decision-maker perceives the level of analysis. If applied at either a national or global level, this tool has the potential to serve as a common language or standard framework that policymakers could use to describe social sustainability.

**Supplementary Materials:** The following supporting information can be downloaded at: https://www.mdpi.com/article/10.3390/su151612107/s1, File S1: Expert Quationnaire AHP; File S2: Nettskjema Questionnaire; File S3: NSD Appovement; File S4: Result of Open Questions in Sureveys.

**Author Contributions:** Conceptualization, T.A.; Methodology, T.A.; Writing—review & editing, A.T.S. and A.J.; Supervision, A.T.S and A.J. All authors have read and agreed to the published version of the manuscript.

**Funding:** This study is part of the ongoing research project "CaPs-Citizens as Pilots of Smart Cities", funded by Nordforsk (project number 95576).

**Institutional Review Board Statement:** Not applicable.

**Informed Consent Statement:** Not applicable.

**Data Availability Statement:** Data supporting this study are included within the article and Supporting Materials.

**Acknowledgments:** Farah Homayun Urban Planner from Alna district in Oslo and Zahir Barahmand.

**Conflicts of Interest:** The authors declare no conflict of interest.

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
