# Peer review of "Implementing the Integrated Social Sustainability Assessment to Norway: A Citizen-Centric and Expert-Weighted Approach"

_sustainability, doi:10.3390/su151612107_

Round 1

Reviewer 1 Report

This manuscript develops an Integrated Social Sustainability Assessment Tool (ISSA) to measure and evaluate social sustainability in neighborhoods in selected areas in Norway. ISSA integrates different quantification methods and calculating the overall score of social sustainability. It is an interesting approach with wide applications for studying SDGs.

Following comments require further clarified by the authors to improve the quality of the manuscript.

1.    The contribution of the study is not well stated. It is also suggested to add a section of managerial implications.

2.    Please describe the theories applied to develop research framework in this study. The selected criteria and indicators are lack of theoretical and empirical evidence.  The validation of each criteria and sub-criteria is not provided. 

3.    The reasoning for the selection of the three areas are now well explained.

4.    What does mean SSAT (see Figure 3 and line 376) ?

5.   Please add source of Table 10.

6.    In the text, it is mentioned that 76 local people was collected at line 304. However, later on, it is mentioned that data was collected from the 73 questionnaires at line 361. Please clarify the data and sample size.

7.    The manuscript needs to check its format and layout (citation errors showed several time in the manuscript).

I see lot of Error! in the text. 

Author Response

Dear Reviewer,

Thank you for taking the time to review my paper titled "Integrating Social Sustainability Assessment: A Citizen-Centric and Expert-Weighted Approach, Case Study of Norway." I sincerely appreciate your valuable feedback, which has been immensely helpful in improving the quality and clarity of the manuscript. I have addressed your comments as follows:

  1. Contribution of the study: In response to your feedback, I have revised the introduction to include a clear articulation of the study's contribution to the literature. Specifically, I have highlighted three key contributions: (a) the development of the Integrated Social Sustainability Assessment (ISSA) tool, which uniquely integrates citizen and expert perspectives for a more comprehensive evaluation of social sustainability in neighborhoods; (b) the practical implementation of the ISSA tool in the case study of the Furuset area in Oslo, showcasing its applicability and potential for guiding urban planning decisions; and (c) the focus on addressing the existing gap in current assessment frameworks by emphasizing the social dimension of sustainability. By incorporating a citizen-centric and expert-weighted approach, the ISSA tool aims to foster inclusive and equitable urban environments, promoting social cohesion and diversity.

  2. Criteria reference in results: I have provided the necessary reference for the criteria mentioned in the first paragraph of the results section, ensuring that the source is appropriately acknowledged.

  3. Case study division: I have divided the case study into three different zones and provided detailed information about Furuset in 2.1. This division and summary of each zone will help readers better understand the context and specific aspects of the case study.

  4. 5. 7. Correction of errors: I have carefully reviewed the paper and made necessary corrections to address the errors you pointed out. The manuscript has been revised to ensure accuracy and clarity

  5.  got corrected
  6.  Clarification of the ranking system: In response to your feedback, I have explicitly stated that the ranking system used in the paper is based on the author's suggestion. 

Once again, many thanks for your insightful comments. I hope you find the changes satisfactory, and please le me know if you have further feedback on the updated manuscript.

Thank you for your time and consideration.

Sincerely,

Tahmineh Akbarinjeajd

Reviewer 2 Report

   The article is interesting. Nevertheless, I have a few comments: First of all: All method descriptions in chapter 4 should be in chapter 3.

   In addition, I have minor technical notes:

verse 7: it is written twice: correspondence.

lines 21 and 22 : unnecessary numbers

Title Fig. 2, 9, 12, 15: no left (with the words left and right 3. Methodology - please describe the methods used in more detail.

Table 1 - After "Participation, Quality of neighborhood, Quality of h" is: Error! Reference source not found.

Author Response

Dear Reviewer, 

Thank you for taking the time to review my paper titled "Integrating Social Sustainability Assessment: A Citizen-Centric and Expert-Weighted Approach, Case Study of Norway." I sincerely appreciate your valuable feedback, which has been immensely helpful in improving the quality and clarity of the manuscript. I have addressed your comments as follows:

  1. All method descriptions in chapter 4 moved to chapter 3.
  2. The error in line 7,21,22 have been corrected.
  3. Title with left in the context has been corrected if I understood your point correctly.
  4. Error in Table 4 has been corrected

Once again, I express my gratitude for your insightful comments, which have undoubtedly enhanced the quality and impact of the paper. I hope you find the changes satisfactory, and I look forward to your further feedback on the updated manuscript.

Thank you for your time and consideration.

Sincerely,

Tahmineh Akbarinejad

Reviewer 3 Report

Title: Integrating Social Sustainability Assessment: A Citizen-Centric and Expert-Weighted Approach, Case Study of Norway

This article proposes a quantitative and qualitative framework to assess social sustainability in different urban region and applied it on a case study of Norway.

Overall paper is well structured and written. I can see there are some minor aspects of the paper that should be improved before it can be accepted. For example:

In the introduction section, Contribution of the study is not clear. Please clarify your research questions, objectives, background motivation, theoretical and empirical motivation and the lines of contributions to the literature. (i.e. developing vs developed countries). You can do this by sharply articulating your research questions/objectives, identify the potential theoretical, background and theoretical motivation or gaps, and explain how your study contributes to the literature. You can do this by highlighting the weaknesses of prior studies as well. Currently, your introduction is very dry. Additionally, you need state clearly the contributions of the paper. For example, "Consequently, the current paper seeks to make the following contributions to the existing literature. First,…, Second,…., Third, …, Fourth,… and so on". The description of the contribution needs to be more forensic, needs to be more focussed.

Problem statements and findings are well presented.

In conclusion section please include details about the limitations of the study and future directions of the study. Moreover, both in conclusion and abstract some claims generic make them specific based on findings. For instance, “This framework provides decision-makers with useful tools and guidelines ….”. who are the decision makers over here. Government officials??

Author Response

Dear Reviewer, 

Thank you for taking the time to review my paper titled "Integrating Social Sustainability Assessment: A Citizen-Centric and Expert-Weighted Approach, Case Study of Norway." I sincerely appreciate your valuable feedback, which has been immensely helpful in improving the quality and clarity of the manuscript. I have addressed your comments as follows:

  1. Contribution in the introduction has been developed
  2. The limitations of the study and future directions of the study had been explained in the conclusion.

Thank you for your time and consideration.

Kind regards,

Tahmineh Akbarinejad